# Mutual Information and Quantum Discord in Quantum State Discrimination with a Fixed Rate of Inconclusive Outcomes

**DOI:** 10.3390/e23010073

**Published:** 2021-01-06

**Authors:** Omar Jiménez, Miguel Angel Solís–Prosser, Leonardo Neves, Aldo Delgado

**Affiliations:** 1Centro de Óptica e Información Cuántica, Facultad de Ciencias, Universidad Mayor, Camino La Pirámide N°5750, Huechuraba 8580745, Chile; 2Departamento de Ciencias Físicas, Universidad de La Frontera, Temuco 4811230, Chile; miguel.solis@ufrontera.cl; 3Departamento de Física, Universidade Federal de Minas Gerais, Belo Horizonte 31270-901, Brazil; lneves@fisica.ufmg.br; 4Departamento de Física, Universidad de Concepción, P.O. BOX 160-C, Concepción 4070043, Chile; aldelgado@udec.cl

**Keywords:** quantum state discrimination, accessible information, quantum discord

## Abstract

We studied the mutual information and quantum discord that Alice and Bob share when Bob implements a discrimination with a fixed rate of inconclusive outcomes (FRIO) onto two pure non-orthogonal quantum states, generated with arbitrary a priori probabilities. FRIO discrimination interpolates between minimum error (ME) and unambiguous state discrimination (UD). ME and UD are well known discrimination protocols with several applications in quantum information theory. FRIO discrimination provides a more general framework where the discrimination process together with its applications can be studied. In this setting, we compared the performance of optimum probability of discrimination, mutual information, and quantum discord. We found that the accessible information is obtained when Bob implements the ME strategy. The most (least) efficient discrimination scheme is ME (UD), from the point of view of correlations that are lost in the initial state and remain in the final state, after Bob’s measurement.

## 1. Introduction

During the last few decades, the classical and quantum correlations [1,2] present in quantum communication protocols have been an important subject of study [3,4,5,6]. In a quantum communication protocol, Alice typically sends information to Bob that is encoded in quantum states [7]. In order to access the information, Bob must implement a quantum measurement. This quantum measurement is, in general, an irreversible process [8] that changes the quantum state, produces decoherence and also entropy [9]. However, Bob could choose a particular quantum measurement in order to optimize some figure of merit, for instance, the mutual information, the Bayes cost, or the quantum correlations, among others [2,10,11]. These quantities allow us to evaluate the performance of a signal-detection process [11], and, in this context, the study of quantum state discrimination strategies becomes crucial.

One of the most important problems in quantum information theory, with application in quantum cryptography [12,13], is to find the accessible information which corresponds to the maximum of mutual information [14,15,16]. This is a hard mathematical problem in which only the results for some lower and upper bounds [11,17,18] are known. The Bayes cost problem is associated with the minimization of an average cost function [19], for example, to find the minimum probability of error in a scheme of quantum state discrimination [20]. In particular, if Alice uses two non-orthogonal pure states, it is known that the minimum error discrimination implemented by Bob maximizes the mutual information and minimizes the error probability [11,16,21]. On the other hand, quantum correlations [3,4,5,6] can be used as a criteria of quantumness of the states [22,23] and also these correlations allow us to quantify the resources that are required to carry out quantum communication protocols [24,25,26]. Some of the most studied quantum correlations are: entanglement [27], quantum discord [2,28], thermal discord [29,30], and global discord [31,32].

If the set of states used by Alice, in the quantum discrimination scenario, contains two or more non-orthogonal states, then it is impossible for Bob to distinguish them deterministically [33]. In the literature, there are several strategies that can be used to distinguish or discriminate between these states [16,20]. The strategy we need to use will depend on the particular application. Two of the most known strategies are the minimum error discrimination (ME) [34,35] and the unambiguous state discrimination (UD) [36,37,38]. In ME, the discrimination of the non-orthogonal states is carried out in such a way that the probability of mistaking a retrodiction is minimized [9,16,33]. In UD, the discrimination of non-orthogonal states is carried out without error, but we must introduce an inconclusive result [39]. These schemes of discrimination are required to implement quantum teleportation [40,41], entanglement swapping [42,43], quantum cryptography [44], dense coding [45], and entanglement concentration [46]. Some of the experimental realizations by these methods can be found in [12,47,48,49] for ME and in [50,51] for UD. The ME and UD schemes can be studied simultaneously in a single protocol of discrimination [52,53]. This protocol is known by the name of fixed rate of inconclusive outcomes (FRIO), where we minimize the error probability in the discrimination of the non-orthogonal states, under the constraint of a fixed rate of the inconclusive outcomes [54,55,56]. As we already said, FRIO interpolates continuously between ME and UD. Another process with this characteristic is parametric separation [57], where a set of pure states is transformed into another set which is more distinguishable. FRIO discrimination and parametric separation allow us to study quantum state discrimination in a more general framework. A complete analysis for FRIO discrimination was done by Bagan et al. [58], in terms of the optimal probabilities in the discrimination for two pure non-orthogonal states with arbitrary a priori probabilities.

Usually, the state discrimination processes are studied only in terms of the optimum probability of success of the protocol [16,20,33]. The aim of this work is to study the success probability, and also the mutual information (MI) that Alice and Bob share as well as the quantum discord (QD) involved in the scheme of FRIO discrimination. This was done for two pure non-orthogonal states with arbitrary a priori probabilities, and it allows us to compare simultaneously the performance of ME, UD, and intermediate cases of FRIO discrimination using the aforementioned quantities. In particular, we consider quantum discord as a quantum correlation because it is directly related with the loss of quantum information produced by FRIO discrimination. This is a generalization of a previous work [9], where we considered only the strategy of ME as a subject of study. In this work, we consider the cases in which Alice and Bob share a separable quantum channel. We found that the most (least) efficient discrimination scheme is ME (UD), from the point of view of QD that is lost in the initial state and the MI that remains in the final state after the measurement.

This article is organized as follows: In Section 2, we review the FRIO discrimination for two non-orthogonal states. In Section 3, we describe the initial and final Alice’s and Bob’s states after the measurement implemented by Bob, according to FRIO. In Section 4 and Section 5, we study the mutual information and the quantum discord, respectively, when Bob implements FRIO discrimination. Finally, in Section 6, we summarize our results and show our conclusions.

## 2. FRIO Discrimination

We consider a scheme in which Alice sends Bob a state from the set of two pure non-orthogonal states $\{|\phi_{1}\rangle ,|\phi_{2}\rangle \}$, generated with arbitrary a priori probabilities $\eta_{1}$ and $\eta_{2}$, respectively, such that $\eta_{1}+\eta_{2}=1$. The overlap between the non-orthogonal states is denoted by $\alpha =\langle \phi_{1}|\phi_{2}\rangle$, which we consider as a real parameter in the interval α∈[0,1]. In this scenario, Bob must implement the discrimination by FRIO. In the scheme of FRIO, the probability of error $P_{e}$ is minimized under the constraint of a fixed rate of inconclusive outcomes *Q* [52,53,54,55,56]. The present study follows the results found in Ref. [58] for the discrimination of two pure non-orthogonal states with arbitrary a priori probabilities. The process of FRIO discrimination is carried out by Bob using a positive operator valued measure (POVM) with three elements, such that
(1)$$\Pi_{1}+\Pi_{2}+\Pi_{0}=𝟙,$$
with 𝟙 being the identity operator on the two-dimensional Hilbert space spanned by the states $\{|\phi_{1}\rangle ,|\phi_{2}\rangle \}$. Here, $\Pi_{1(2)}$ corresponds to the element of POVM that identifies $|\phi_{1(2)}\rangle$ and $\Pi_{0}$ corresponds to the operator associated with the inconclusive outcomes [58]. The average probabilities of success $P_{s}$, error $P_{e}$, and inconclusive outcomes *Q* are given by
(2)$$\begin{array}{ccc} P_{s} & = & \mathrm{tr}\left(\eta_{1}\rho_{1}\Pi_{1}\right)+\mathrm{tr}\left(\eta_{2}\rho_{2}\Pi_{2}\right)=\eta_{1}p_{1}+\eta_{2}p_{2}, \end{array}$$
(3)$$\begin{array}{ccc} P_{e} & = & \mathrm{tr}\left(\eta_{1}\rho_{1}\Pi_{2}\right)+\mathrm{tr}\left(\eta_{2}\rho_{2}\Pi_{1}\right)=\eta_{1}r_{1}+\eta_{2}r_{2}, \end{array}$$
(4)$$\begin{array}{ccc} Q & = & \mathrm{tr}\left(\rho \Pi_{0}\right)=\eta_{1}q_{1}+\eta_{2}q_{2}, \end{array}$$
where $\rho_{1(2)}=|\phi_{1(2)}\rangle \langle \phi_{1(2)}|$ and $\rho =\eta_{1}\rho_{1}+\eta_{2}\rho_{2}$. Here, $p_{1(2)}$, $r_{1(2)}$, and $q_{1(2)}$ represent the probabilities of success, error, and inconclusive outcomes in the discrimination of non-orthogonal states $|\phi_{1(2)}\rangle$, respectively. These probabilities satisfy the following conditions:(5)$$p_{i}+r_{i}+q_{i}=1,\;\mathrm{for}\;i=1,2,$$
which implies that the average probabilities, in Equations (Equation 2)–(Equation 4), meet the condition
(6)$$P_{s}+P_{e}+Q=1.$$

The optimal strategy in the FRIO discrimination minimizes $P_{e}$ under the constraint that *Q* is fixed. Given the symmetry, we consider the cases where $\eta_{1}\leq \eta_{2}$, or, equivalently, $0\leq \eta_{1}\leq 1/2$. This implies that the optimal probabilities, for inconclusive outcomes, error, and success, are given respectively by [58,59]
(7)$$\begin{array}{ccc} q_{i} & = & \frac{Q}{2\eta_{i}}, \end{array}$$
(8)$$\begin{array}{ccc} r_{i} & = & \frac{1}{2}\left(1-q_{i}-\frac{\left(1-q_{i}\right)\bar{Q}-\frac{{\left(Q_{0}-Q\right)}^{2}}{2\eta_{i}}}{\sqrt{{\bar{Q}}^{2}-{\left(Q_{0}-Q\right)}^{2}}}\right), \end{array}$$
(9)$$\begin{array}{ccc} p_{i} & = & 1-q_{i}-r_{i}, \end{array}$$
with $\bar{Q}=1-Q$ and $Q_{0}=2\sqrt{\eta_{1}\eta_{2}}\alpha$. From these expressions, the optimal average error probability $P_{e}$ in Equation (3) is
(10)$$P_{e}=\frac{1}{2}\left(\bar{Q}-\sqrt{{\bar{Q}}^{2}-{\left(Q_{0}-Q\right)}^{2}}\right).$$

Equations (Equation 7)–(Equation 10) are valid in the intervals I and II, where the interval I is defined by
(11)$$\begin{array}{ccc} \frac{\alpha^{2}}{1+\alpha^{2}}\leq & \eta_{1} & \leq 1/2, \end{array}$$
(12)$$\begin{array}{ccc} 0\leq & Q & \leq Q_{0}, \end{array}$$
and the interval II is defined by
(13)$$\begin{array}{ccc} 0\leq & \eta_{1} & \leq \frac{\alpha^{2}}{1+\alpha^{2}}, \end{array}$$
(14)$$\begin{array}{ccc} 0\leq & Q & \leq Q_{th}, \end{array}$$
where the threshold rate $Q_{th}=\frac{2\eta_{1}\eta_{2}\left(1-\alpha^{2}\right)}{1-Q_{0}}$ separates the intervals II and III. In the interval III, defined by
(15)$$\begin{array}{ccc} 0\leq & \eta_{1} & \leq \frac{\alpha^{2}}{1+\alpha^{2}}, \end{array}$$
(16)$$\begin{array}{ccc} Q_{th}\leq & Q & \leq \eta_{1}+\eta_{2}\alpha^{2}, \end{array}$$
the optimal FRIO discrimination is obtained when we consider the following expressions: (17)$$\begin{array}{ccc} r_{1} & = & \frac{P_{e}}{\eta_{1}},\;p_{1}=0,\;q_{1}=1-r_{1}, \end{array}$$(18)$$\begin{array}{ccc} r_{2} & = & 0,\;p_{2}={\left[\alpha \sqrt{r_{1}}+\sqrt{\left(1-r_{1}\right)\left(1-\alpha^{2}\right)}\right]}^{2},\;q_{2}=1-p_{2}. \end{array}$$

In interval III, the average probability of error $P_{e}$ is given by
(19)$$P_{e}=\frac{1}{1-4c}\left(\eta_{1}\bar{Q}+c\left(1-2\eta_{1}-2\bar{Q}\right)-Q_{0}\sqrt{c(Q\bar{Q}-c)}\right),$$
where $c=\eta_{1}\eta_{2}\left(1-\alpha^{2}\right)$. In interval III, the optimum strategy is a two-element projective measurement [58]. The FRIO discrimination interpolates between ME discrimination (Q=0) and UD when *Q* takes its maximum possible value (given by $Q_{max}=Q_{0}$ in the interval I and $Q_{max}=\eta_{1}+\eta_{2}\alpha^{2}$ in the interval III). In our analysis, we considered two intermediate cases for the rate of inconclusive outcomes *Q*, which are $Q=Q_{0}/3$ and $Q=2Q_{0}/3$. The three intervals and the four cases for *Q* are shown in Figure 1. The intervals I, II, and III are associated with the regions in yellow, gray, and cyan, respectively. On the other hand, the cases of $Q=Q_{max}$, $Q=2Q_{0}/3$, $Q=Q_{0}/3$, and Q=0, which appear in all the figures throughout this work, are associated with a dashed black line, a dashed-dotted blue line, a dotted red line, and a solid green line, respectively. In the scheme of FRIO discrimination, we have three free parameters which are the overlap between the non-orthogonal states α, the a priori probability $\eta_{1}$, and the rate of inconclusive outcomes *Q*. In the following, we have identified the case Q=0 with the ME and the case $Q=Q_{max}$ with UD.

Figure 1 shows the intervals I, II, and III defined above for three fixed values of $\eta_{1}$. We see that the region size of each interval depends on the value of the parameter $\eta_{1}$. For instance, if the value of $\eta_{1}=0.5$$\left(\eta_{1}=0\right)$, only interval I (III) exists. For other values of $\eta_{1}$, there are three intervals as shown in Figure 1b,c. Moreover, from Figure 1, it is clear that, when the states are orthogonal (α=0), there is a single value possible for *Q* which is Q=0. On the other hand, if the states are equal (α=1), the value of *Q* can assume any value in the range 0≤Q≤1.

Figure 2 shows the success probability $P_{s}$ (Equation (Equation 2)) in the discrimination of the non-orthogonal states $\{|\phi_{1}\rangle ,|\phi_{2}\rangle \}$ as a function of α for several values of *Q* and $\eta_{1}$. The success probability is defined in the three intervals and can be obtained from Equation (Equation 6), i.e., $P_{s}=1-P_{e}-Q$. For a fixed value of *Q* and $\eta_{1}$, the maximum value of success probability is $P_{s}=1$, when the states $\{|\phi_{1}\rangle ,|\phi_{2}\rangle \}$ are orthogonal (α=0). If we increase the value of α, the success probability decreases until its minimum value which depends on *Q* and $\eta_{1}$. In general, for any value of α and $\eta_{1}$, the biggest value of $P_{s}$ is obtained when Q=0, which corresponds to the case of ME. If we increase the rate of inconclusive outcomes, the success probability of discrimination decreases until it adopts its minimum value when $Q=Q_{max}$, which is associated with the case of UD. Moreover, for a fixed value of *Q* and α, the success probability is higher when we decrease the value of $\eta_{1}$.

In order to implement FRIO discrimination, we need to increase the initial two-dimensional Hilbert space H spanned by the states $\{|\phi_{1}\rangle ,|\phi_{2}\rangle \}$. This can be done by the method of direct sum extension [60], K=H⊕A, where A represents a one-dimensional ancillary subspace, spanned by the |0〉 state. We assume that there is a unitary transformation *U* acting on K in such a way that it generates the following transformation [59,60,61]:(20)$$\begin{array}{ccc} U|\phi_{1}\rangle & = & \sqrt{p_{1}}|1\rangle +\sqrt{r_{1}}|2\rangle +\sqrt{q_{1}}|0\rangle , \end{array}$$(21)$$\begin{array}{ccc} U|\phi_{2}\rangle & = & \sqrt{r_{2}}|1\rangle +\sqrt{p_{2}}|2\rangle +\sqrt{q_{2}}|0\rangle . \end{array}$$

Therefore, the elements of the POVM for implementing FRIO discrimination are rank-one orthogonal projections $\{\Pi_{i}=|i\rangle \langle i|\}$ in a three-dimensional Hilbert space. The inconclusive result, $\Pi_{0}$, is associated with the state |0〉 of the ancilla. In this case, the non-orthogonal states $\{|\phi_{1}\rangle ,|\phi_{2}\rangle \}$ are projected to the same state |0〉 with probabilities $\left\{q_{1},q_{2}\right\}$, respectively. On the other hand, the POVM elements $\Pi_{1(2)}$ are associated with the detection of the states |1(2)〉 in the original Hilbert space. In this form, the projection in |1(2)〉 is associated with success (failure), with probabilities $p_{1}\left(r_{1}\right)$, in the discrimination of the state $|\phi_{1}\rangle$ and with failure (success), with probabilities $r_{2}\left(p_{2}\right)$, in the discrimination of the state $|\phi_{2}\rangle$. In this way, when we consider the probabilities given by Equations (Equation 7)–(Equation 9) for the intervals *I* and II, and Equations (Equation 17) and (Equation 18) for the interval III, we are implementing the FRIO discrimination. Given that *U* in Equations (Equation 20) and (Equation 21) is a unitary transformation, the respective probabilities must satisfy the following condition:(22)$$\alpha =\sqrt{p_{1}r_{2}}+\sqrt{r_{1}p_{2}}+\sqrt{q_{1}q_{2}}.$$

In expressions of Equations (Equation 20)–(Equation 22), for instance, implementation of ME discrimination requires one to consider $q_{1}=q_{2}=0$, whereas implementation of UD demands one to take $r_{1}=r_{2}=0$.

## 3. Channel without Entanglement

As it was done in Ref. [9], let us consider that the communicating parties, Alice and Bob, initially share the following separable joint quantum state:(23)$$\rho_{AB}=\sum_{i=1}^{2}\eta_{i}{|i\rangle }_{A}\langle i|⊗|\phi_{i}\rangle_{B}\langle \phi_{i}|,$$
where states ${\{|1\rangle }_{A}{,|2\rangle }_{A}\}$ form an orthonormal base for Alice’s two-dimensional quantum system, and ${\{|\phi_{1}\rangle }_{B},|\phi_{2}\rangle_{B}\}$ are two non-orthogonal states of Bob’s quantum system, given by
(24)$$\begin{array}{c} |\phi_{1}\rangle_{B}={a|1\rangle }_{B}+b{|2\rangle }_{B}, \end{array}$$
(25)$$\begin{array}{c} |\phi_{2}\rangle_{B}={b|1\rangle }_{B}+a{|2\rangle }_{B}. \end{array}$$

From the normalization condition, the real coefficients *a* and *b* satisfy, $a^{2}+b^{2}=1$, and the overlap between the non-orthogonal states is $\alpha =\langle \phi_{1}|\phi_{2}\rangle =2ab$. We consider, for simplicity, that $b\in [0,1/\sqrt{2}]$ and therefore α∈[0,1]. Alice prepares a single copy of a quantum system in the state $|\phi_{i}\rangle_{B}$ and sends it to Bob with an a priori probability $\eta_{i}$. Thereby, Alice and Bob share quantum and classical correlations encoded in the joint state $\rho_{AB}$ of Equation (Equation 23). Alice’s and Bob’s initial states $\rho_{A}$ and $\rho_{B}$, prior to the application of any transformation or measurement, are $\rho_{A}={\mathrm{tr}}_{B}\left(\rho_{AB}\right)$ and $\rho_{B}={\mathrm{tr}}_{A}\left(\rho_{AB}\right)$, respectively, where
(26)$$\begin{array}{ccc} \rho_{A} & = & \sum_{i=1}^{2}\eta_{i}{|i\rangle }_{A}\langle i|, \end{array}$$
(27)$$\begin{array}{ccc} \rho_{B} & = & \sum_{i=1}^{2}\eta_{i}|\phi_{i}\rangle_{B}\langle \phi_{i}|. \end{array}$$

Here, we consider that, once Bob has received the single copy of the quantum system in the state $|\phi_{i}\rangle_{B}$, he implements the FRIO discrimination. Next, we studied the mutual information that Alice and Bob share, and the quantum discord or the quantum correlation that are lost when Bob implements FRIO discrimination onto his states.

To implement FRIO discrimination, Bob first applies the unitary transformation $U_{B}$, given by Equations (Equation 20) and (Equation 21), onto his quantum system. Thereby, the initial joint state $\rho_{AB}$ between Alice and Bob of Equation (Equation 23) changes to ${\hat{\rho }}_{AB}=\left({𝟙}_{A}⊗U_{B}\right)\rho_{AB}\left({𝟙}_{A}⊗U_{B}^{†}\right)$, where
(28)$${\hat{\rho }}_{AB}=\sum_{i=1}^{2}\eta_{i}{|i\rangle }_{A}\langle i|⊗|\hat{\phi_{i}}\rangle_{B}\langle \hat{\phi_{i}}|,$$
with $|\hat{\phi_{i}}\rangle_{B}=U_{B}{|\phi_{i}\rangle }_{B}$. The unitary transformation $U_{B}$ of Equations (Equation 20) and (Equation 21), applied by Bob onto his quantum system, is a reversible process [8]. Therefore, the quantum correlations between Alice and Bob are not changed by this transformation. Now, we assume that Bob can implement his measurement with an arbitrary POVM $\{\Pi_{b}^{i}=|\psi_{i}\rangle \langle \psi_{i}|\}$ in a three-dimensional Hilbert space spanned by the basis $\left\{|\psi_{i}\rangle \right\}$. This basis is defined by the following states [62,63]: (29)$$\begin{array}{ccc} |\psi_{1}\rangle & = & e^{i\rho }[c_{1}c_{2}|1\rangle +s_{1}e^{i\delta }|2\rangle +c_{1}s_{2}e^{i\beta }|0\rangle ], \end{array}$$(30)$$\begin{array}{ccc} |\psi_{2}\rangle & = & e^{i\sigma }[-\left(s_{1}c_{2}c_{3}e^{-i\delta }+s_{2}s_{3}e^{i(\gamma -\beta )}\right)|1\rangle +c_{1}c_{3}|2\rangle +\left(c_{2}s_{3}e^{i\gamma }-s_{1}s_{2}c_{3}e^{\beta -\delta }\right)|0\rangle ], \end{array}$$(31)$$\begin{array}{ccc} |\psi_{0}\rangle & = & e^{i\tau }[\left(s_{1}c_{2}s_{3}e^{-i(\delta +\gamma )}-s_{2}c_{3}e^{-i\beta }\right)|1\rangle -c_{1}s_{3}e^{-i\gamma }|2\rangle +\left(c_{2}c_{3}+s_{1}s_{2}s_{3}e^{i(\beta -\delta -\gamma )}\right)|0\rangle ], \end{array}$$
where $c_{i}=\cos\theta_{i}$ and $s_{i}=\sin\theta_{i}$. In particular, FRIO discrimination corresponds to the case where the measurement basis is given by {|1〉,|2〉,|0〉}, that is, for $c_{i}=1$ and all the phases in Equations (Equation 29)–(Equation 31) are equal to zero, which means that $\left\{|\psi_{i}\rangle =|i\rangle \right\}$. The measurement carried out by Bob on his quantum system generates three conditional post-measurement states $\rho_{A|b}^{i}$ for Alice’s quantum system. If Bob found the state $|\psi_{i}\rangle$ with probability $p_{i}^{b}=\mathrm{tr}\left(\Pi_{b}^{i}{\hat{\rho }}_{AB}\right)$, Alice’s conditional state will be $\rho_{A|b}^{i}={\mathrm{tr}}_{B}\left(\Pi_{b}^{i}{\hat{\rho }}_{AB}\right)/p_{i}^{b}$. Therefore, Alice’s post-measurement states are
(32)$$\begin{array}{c} \rho_{A|b}^{i}=\frac{1}{p_{i}^{b}}\left(\right|t_{i1}{|}^{2}{|1\rangle }_{A}\langle 1|+|t_{i2}{|}^{2}{|2\rangle }_{A}\langle 2|),\;\mathrm{for}\;i=0,1,2, \end{array}$$
where
(33)$$\begin{array}{ccc} |t_{ij}{|}^{2} & = & \eta_{j}|\langle \psi_{i}|U_{B}|\phi_{j}{\rangle |}^{2},\;\mathrm{for}\;j=1,2, \end{array}$$
(34)$$\begin{array}{ccc} p_{i}^{b} & = & |t_{i1}{|}^{2}+{\left|t_{i2}\right|}^{2}, \end{array}$$
with
(35)$$\sum_{i=0}^{2}p_{i}^{b}=1.$$

The quantum measurement implemented by Bob, in general, changes the initial joint state from $\rho_{AB}$ in a composite Hilbert space of 2⊗2 to the final average joint state $\rho_{AB}^{\prime }$ in a Hilbert space of 2⊗3, which is given by [9,28]
(36)$$\rho_{AB}^{\prime }=\sum_{i=0}^{2}p_{i}^{b}\rho_{A|b}^{i}⊗\Pi_{b}^{i},$$
where $\Pi_{b}^{i}$ are the projectors $|\psi_{i}\rangle_{B}\langle \psi_{i}|$ in the three-dimensional Hilbert space K. The final joint state $\rho_{AB}^{\prime }$ in Equation (Equation 36) is a classical state because there are local measurements in Alice’s and Bob’s systems that do not perturb it [28]. The average final reduced states for Alice’s and Bob’s quantum systems are given by
(37)$$\begin{array}{ccc} \rho_{A}^{\prime } & = & \sum_{i=0}^{2}p_{i}^{b}\rho_{A|b}^{i}, \end{array}$$
(38)$$\begin{array}{ccc} \rho_{B}^{\prime } & = & \sum_{i=0}^{2}p_{i}^{b}|\psi_{i}\rangle_{B}\langle \psi_{i}|. \end{array}$$

Then, the final reduced state for Alice’s system does not change because
(39)$$\sum_{i=0}^{2}{\left|t_{ij}\right|}^{2}=\eta_{j}.$$

This implies that $\rho_{A}^{\prime }=\eta_{1}{|1\rangle }_{A}\langle 1|+\eta_{2}{|2\rangle }_{A}\langle 2|=\rho_{A}$. On the other hand, in general, the reduced state for Bob’s system changes.

## 4. Mutual Information

In a bipartite state $\rho_{AB}$, the total amount of correlations, in the many copies scenario [64], is given by the quantum mutual information, which is defined as [1,2,28,64]
(40)$$I\left(\rho_{AB}\right)=S\left(\rho_{A}\right)+S\left(\rho_{B}\right)-S\left(\rho_{AB}\right),$$
where S(ρ) is the von Neumann entropy of the state ρ, given by $S\left(\rho \right)=-\sum_{i}\lambda_{i}\log_{2}\lambda_{i}$, where $\lambda_{i}$ are the eigenvalues of ρ. Hence, in the protocol of FRIO discrimination, we consider that Alice emits many copies of independent identically distributed (i.i.d.) data. In this scenario, the global quantum state is given by $\sigma =\rho_{AB}^{⊗n}$ for some large *n* [9,65] and Bob implements his measurement on each single copy that he has, using a POVM with elements $\{\Pi_{b}^{i}=|\psi_{i}\rangle \langle \psi_{i}|\}$. The entropy of the joint initial state $\rho_{AB}$, given by Equation (Equation 23), is $S\left(\rho_{AB}\right)=S\left(\rho_{A}\right)+\sum \eta_{i}S{(|\phi_{i}\rangle }_{B}\langle \phi_{i}|)=S\left(\rho_{A}\right)$. Therefore, the initial quantum mutual information between Alice and Bob is equal to $I\left(\rho_{AB}\right)=S\left(\rho_{B}\right)$, where the eigenvalues of Bob’s states $\rho_{B}$ are
(41)$$\lambda_{1}^{b}=\frac{1}{2}\left(1+\sqrt{1-4\eta_{1}\eta_{2}\left(1-\alpha^{2}\right)}\right),$$
and $\lambda_{2}^{b}=1-\lambda_{1}^{b}$. The quantum mutual quantum information can be written as [2,28]
(42)$$I\left(\rho_{AB}\right)=J\left(A|\left\{\Pi_{b}\right\}\right)+D\left(A|\left\{\Pi_{b}\right\}\right),$$
where $J(A|\left\{\Pi_{b}\right\})$ are the classical correlations or the classical mutual information between Alice and Bob and $D(A|\left\{\Pi_{b}\right\})$ is the quantum discord. These two quantities depend on the measurement implemented by Bob, with the POVM $\left\{\Pi_{b}\right\}$, but their sum does not [5], i.e., they are complementary to each other. The classical correlations $J(A|\left\{\Pi_{b}\right\})$ between Alice and Bob are defined as [1,28]
(43)$$J\left(A|\left\{\Pi_{b}\right\}\right)=S\left(\rho_{A}\right)-\sum_{i=0}^{2}p_{i}^{b}S\left(\rho_{A|b}^{i}\right),$$
which is the information about Alice’s system gained by Bob by means of the measurement $\left\{\Pi_{b}\right\}$ [28]. The accessible information to Bob [10,15] is the maximal classical correlation $J(A|\left\{\Pi_{b}\right\})$ with respect to all possible measurements implemented by Bob, where
(44)$$J\left(A|B\right)=\max_{\left\{\Pi_{b}\right\}}J\left(A|\left\{\Pi_{b}\right\}\right)=S\left(\rho_{A}\right)-\min_{\left\{\Pi_{b}\right\}}\sum_{i=0}^{2}p_{i}^{b}S\left(\rho_{A|b}^{i}\right).$$

Then, the expression J(A|B) represents the classical mutual information maximized with respect to the detection strategy [12]. To determine the J(A|B), without doing the maximization in Equation (Equation 44), we use the Koashi–Winter relation [66]. This can be done because the state of Equation (Equation 23) can be purified, using an auxiliary third quantum system *C*, in the following form:(45)$${|\Psi \rangle }_{ABC}=\sum_{i=1}^{2}\sqrt{\eta_{i}}{|i\rangle }_{A}⊗|\phi_{i}\rangle_{B}⊗{|i\rangle }_{C}.$$

Therefore, the maximum classical mutual information between Alice and Bob, when Bob implements his measurement, can be obtained from
(46)$$J\left(A|B\right)=S\left(\rho_{A}\right)-E_{f}\left({\bar{\rho }}_{AC}\right),$$
where, if we denote $\bar{\rho }={|\Psi \rangle }_{ABC}\langle \Psi |$, then ${\bar{\rho }}_{AC}={\mathrm{tr}}_{B}\bar{\rho }$ and the entanglement of formation [67] between subsystems *A* and *C* is given by
(47)$$E_{f}\left({\bar{\rho }}_{AC}\right)=-\lambda_{+}\log_{2}\lambda_{+}-\lambda_{-}\log_{2}\lambda_{-},$$
where
(48)$$\lambda_{\pm }=\frac{1}{2}\left(1\pm \sqrt{1-4\eta_{1}\eta_{2}\alpha^{2}}\right).$$

In our case, accessible information J(A|B) is obtained when Bob implements the strategy of ME discrimination [9,16,20,21], using a two-dimensional projective measurement in the basis {|1〉,|2〉}. This means that the rate of inconclusive outcomes is zero, Q=0, and, thereby, the rate of inconclusive outcomes for the $\{|\phi_{1}\rangle ,|\phi_{2}\rangle \}$ states is also zero, i.e., $q_{1}=q_{2}=0$. However, in general, in order to implement any case of FRIO discrimination, we must consider that $q_{1}\neq 0$ and $q_{2}\neq 0$, and Bob must implement the POVM with the projections onto the states {|0〉,|1〉,|2〉}. This implies that the mutual information $J(A|\left\{\Pi_{b}\right\})$ will be lower than the optimal J(A|B), if a POVM different from the ME measurement is implemented.

Figure 3 shows the mutual information between Alice and Bob, encoded in the joint state $\rho_{AB}$ of Equation (Equation 23), as a function of α for some values of *Q* and $\eta_{1}$. Alice and Bob share the maximal mutual information available from any ensemble of quantum states when $\eta_{1}=1/2$ and Q=0, i.e., when Bob implements ME [16,20,21]. We see that Alice’s and Bob’s system are completely classically correlated, i.e, J(A|B)=1 for $\eta_{1}=0.5$ and when the states ${\{|\phi_{1}\rangle }_{B},|\phi_{2}\rangle_{B}\}$ are orthogonal (α=0). If we increase the overlap α between these states, the mutual information decreases for any value of *Q* and $\eta_{1}$. Furthermore, their systems become completely classically uncorrelated, J(A|B)=0, when the overlap is α=1, for any value of *Q* and $\eta_{1}$. From Figure 3, we see that the parameter $\eta_{1}$, for a fixed value of α and *Q*, considerably affects the amount of MI that are shared by Alice and Bob. For a fixed value of α and $\eta_{1}$, the best strategy of FRIO, from the point of view of the MI, corresponds to the implementation of ME, which is the case of FRIO with Q=0. On the other hand, if we consider a fixed value of α and $\eta_{1}$ and we increase the value of *Q*, the mutual information $J(A|\left\{\Pi_{b}\right\})$ decreases. In this case, the minimum of mutual information occurs when we implement UD, i.e., when the rate of inconclusive outcomes takes its maximum value $Q=Q_{max}$. Therefore, for any value of $\eta_{1}$, the MI, in the scheme of FRIO, are limited by the cases of ME (Q=0) and UD ($Q=Q_{max}$). The maximum difference of the mutual information in this two border cases is approximately 0.15 bit.

Figure 4 shows the mutual information between Alice and Bob as a function of success probability for some values of *Q* and $\eta_{1}$. For a fixed value of *Q* and $\eta_{1}$, if we increase the success probability $P_{s}$, the MI increase. We see from Figure 4 that, even though the $P_{s}$ can take its maximum value $P_{s}=1$, for any value of *Q* and $\eta_{1}$, the MI will be $J(A|\left\{\Pi_{b}\right\})=1$ only if $P_{s}=1$ and $\eta_{1}=0.5$ as shown in Figure 3. The accessible information J(A|B) is obtained when Bob implements ME (Q=0), as previously stated. If we increase the value of *Q*, and fix the value of $\eta_{1}$ and the value of MI, the plots go to lower values of success probabilities.

In general, Bob’s choice of which discrimination process will be implemented will depend on the application that he wants to carry out. For instance, in our scenario, if he is interested in implementing quantum cryptography, then he must discriminate the states by ME. On the other hand, if he wants to do perfect quantum teleportation using a non-maximally entangled quantum channel, with the highest possible probability, then he must implement UD [57]. FRIO discrimination can be relevant in the case where we expect to optimize a certain discrimination protocol in terms of more than one variable—for example, the success probability and the average fidelity in a scheme of quantum teleportation using a non-maximally entangled quantum channel [57].

## 5. Quantum Discord

Quantum discord is defined as the difference between the total correlations $I(\rho_{AB})$ and the classical correlations J(A|B) [28], i.e.,
(49)$$D\left(A|B\right)=I\left(\rho_{AB}\right)-J\left(A|B\right).$$

In our case, we have that $I\left(\rho_{AB}\right)=S\left(\rho_{B}\right)$ and J(A|B) is given by Equation (Equation 46). Therefore, quantum discord can be analytically evaluated, and this quantifies the minimum of the quantum correlations that are consumed in the process. In an equivalent way, the quantum discord can be written as
(50)$$D\left(A|B\right)=S\left(\rho_{B}\right)+\min_{\left\{\Pi_{b}\right\}}\sum_{i=0}^{2}p_{i}^{b}S\left(\rho_{A|b}^{i}\right)-S\left(\rho_{AB}\right).$$

The measurement-dependent version of the quantum discord (QDD) is
(51)$$D\left(A|\left\{\Pi_{b}\right\}\right)=S\left(\rho_{B}\right)+\sum_{i=0}^{2}p_{i}^{b}S\left(\rho_{A|b}^{i}\right)-S\left(\rho_{AB}\right).$$

Here, QDD quantifies the quantum correlations that are consumed in the process of FRIO discrimination. This function depends on the POVM $\{\Pi_{b}^{i}=|\psi_{i}\rangle \langle \psi_{i}|\}$ used to implement the measurement. In our case, these measurement operators correspond to the POVM necessary for implementing FRIO discrimination, that is, $\{\Pi_{b}^{i}=|i\rangle \langle i|\}$ with i=0,1,2. Therefore, the QDD can also be analytically evaluated.

Figure 5 shows the quantum discord D(A|B) as a solid green line. It corresponds to the case when Q=0, or equivalently to the case when we implement ME. Figure 5 also shows the measurement-dependent quantum discord $D(A|\left\{\Pi_{b}\right\})$ as a function of *Q* and α for three different values of $\eta_{1}$. There are two particular cases in which these functions are equal to zero, i.e., $D\left(A|B\right)=D(A|\left\{\Pi_{b}\right\})=0$, for any value of *Q* and $\eta_{1}$. This happens when the states ${\{|\phi_{1}\rangle }_{B},|\phi_{2}\rangle_{B}\}$ are orthogonal (α=0), or when their overlap is equal to one. In these cases, Bob’s measurement does not change the joint state $\rho_{AB}^{\prime }={\hat{\rho }}_{AB}$; therefore, the states ${\{|\phi_{1}\rangle }_{B},|\phi_{2}\rangle_{B}\}$ behave like classical states. Moreover, quantum discord, in this case for Q=0, can be used as a measure of quantumness [22,23]. As we already stated in Ref. [9], the maximum value of quantum discord, i.e., for Q=0, occurs when $\alpha =1/\sqrt{2}$ for any value of $\eta_{1}$. This means that the pair of quantum states ${\{|\phi_{1}\rangle }_{B},|\phi_{2}\rangle_{B}\}$ with overlap $\alpha =1/\sqrt{2}$ will have the maximum quantumness possible.

If we increase the value of *Q*, the quantum discord lost in the process, increases, for fixed values of α and $\eta_{1}$. Then, the maximum loss of quantum correlations happens when Bob implements UD onto the states ${\{|\phi_{1}\rangle }_{B},|\phi_{2}\rangle_{B}\}$. Therefore, the most (least) efficient discrimination scheme, from the point of view of quantum correlations that are lost in $\rho_{AB}$ and the mutual information that remain in the final state $\rho_{AB}^{\prime }$, is the scheme of ME (UD). There is a compromise between the error in the state discrimination and the loss of correlations, that is, if we want to minimize the error in the state discrimination, the loss of correlations increases. On the other hand, for fixed values of α and *Q*, if we increase the value of parameter $\eta_{1}$, from $\eta_{1}=0$ to $\eta_{1}=0.5$, the loss of quantum correlations also increases. However, at the same time, the states ${\{|\phi_{1}\rangle }_{B},|\phi_{2}\rangle_{B}\}$ carry more mutual information, as shown in Figure 3.

## 6. Conclusions

We studied the success probability, the mutual information that Alice and Bob share, and quantum discord involved in a quantum state discrimination protocol. These quantities are useful to evaluate the performance of the detection process, and also have several applications in quantum information theory. Our work was carried out with a quantum bipartite state without entanglement, for the case where Bob implements FRIO discrimination onto two pure non-orthogonal quantum states generated with arbitrary a priori probabilities. In our analysis, for each one of the three aforementioned quantities, there are three free parameters that can be modified: the overlap between the non-orthogonal states α, the a priori probability $\eta_{1}$, and the rate of inconclusive outcomes *Q*. The ME (UD) corresponds to the case when Q=0
$\left(Q=Q_{max}\right)$. Additionally, two intermediate cases were considered for $Q=Q_{0}/3$ and $Q=2Q_{0}/3$.

In this work, known results were recovered. For instance, the implementation of ME (Q=0) with $\eta_{1}=0.5$ allows us to obtain the maximal mutual information available from any ensemble of quantum states and the accessible information for any value of $\eta_{1}$ [16,20,21]. The biggest probability of success $P_{s}$ happens when ME is implemented. Moreover, the maximum amount of mutual information is equal to $J\left(A|B\right)=S\left(\rho_{A}\right)$ when the states $\{|\phi_{1}\rangle ,|\phi_{2}\rangle \}$ are orthogonal. Quantum discord is zero when the joint state is classical (α=0 or α=1), and it reaches its maximum when the overlap of the non-orthogonal states is $\alpha =1/\sqrt{2}$ [9].

We explicitly show the behavior of success probability and we found the upper and lower limits for mutual information and quantum discord, as a function of the parameters α, $\eta_{1}$ and *Q* in the FRIO discrimination scheme. These findings help us to understand how these quantities change as a function of the parameters involved in the discrimination process. For the particular case analyzed in this study, the process of minimum error discrimination optimizes success probability, mutual information, and quantum discord, simultaneously. On the other hand, for unambiguous discrimination, the behavior of these quantities is quite the opposite. This implies that the discrimination without error produces a major decrease in mutual information between Alice and Bob; therefore, a biggest cost in terms of quantum correlations, which are quantified by quantum discord lost in the discrimination process.

In general, FRIO discrimination provides a more general framework in which the discrimination process can be studied, where it is possible to analyze the performance of a discrimination protocol with a fixed rate of inconclusive outcomes. This characteristic can be relevant in the scenario where we expect to optimize a certain discrimination protocol in terms of more than one variable. For instance, the success probability and the average fidelity in a scheme of quantum teleportation using a non-maximally entangled quantum channel [57].

As a next step, we plan to perform the same analysis for *n* pure non-orthogonal states considering the FRIO discrimination scheme.

## Figures and Tables

**Figure 1 entropy-23-00073-f001:**
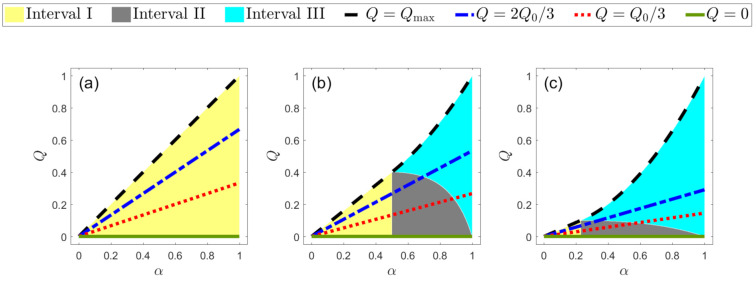
Intervals in FRIO discrimination as a function of α and *Q*, for: (**a**) $\eta_{1}=0.5$, (**b**) $\eta_{1}=0.2$ and (**c**) $\eta_{1}=0.05$. Intervals I, II, and III are represented by the yellow, gray, and cyan regions, respectively. The rate of inconclusive outcomes $Q=Q_{max}$ is the dashed black line, $Q=2Q_{0}/3$ is the dashed-dotted blue line, $Q=Q_{0}/3$ is the dotted red line, and Q=0 is the solid green line.

**Figure 2 entropy-23-00073-f002:**
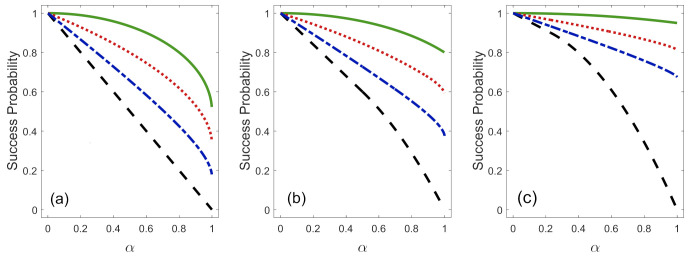
The success probability $P_{s}$ as a function of the overlap α in the discrimination by: ME Q=0 (solid green line), FRIO with $Q=Q_{0}/3$ (dotted red line), FRIO with $Q=2Q_{0}/3$ (dashed-dotted blue line) and UD $Q=Q_{max}$ (dashed black line) for: (**a**) $\eta_{1}=0.5$, (**b**) $\eta_{1}=0.2$, and (**c**) $\eta_{1}=0.05$.

**Figure 3 entropy-23-00073-f003:**
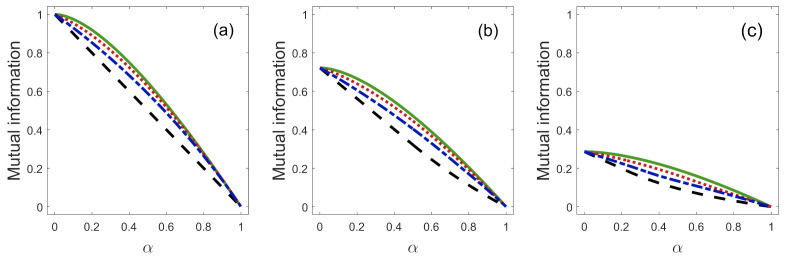
Mutual information between Alice and Bob, $J(A|\left\{\Pi_{b}\right\})$, as a function of the overlap α in the discrimination by: ME where Q=0 (solid green line), FRIO with $Q=Q_{0}/3$ (dotted red line), FRIO with $Q=2Q_{0}/3$ (dashed-dotted blue blue line), and UD where $Q=Q_{max}$ (dashed black line) for: (**a**) $\eta_{1}=0.5$, (**b**) $\eta_{1}=0.2$, and (**c**) $\eta_{1}=0.05$.

**Figure 4 entropy-23-00073-f004:**
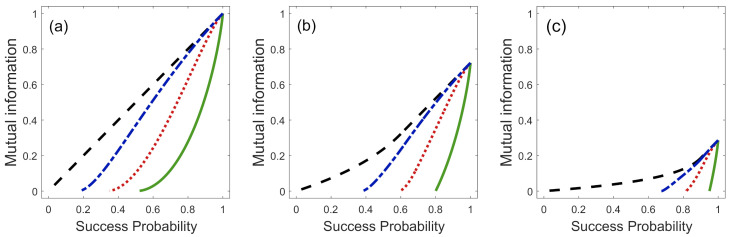
Mutual information as a function of the success probability in the discrimination by: ME where Q=0 (solid green line), FRIO with $Q=Q_{0}/3$ (dotted red line), FRIO with $Q=2Q_{0}/3$ (dashed-dotted blue line) and UD where $Q=Q_{max}$ (dashed black line) for: (**a**) $\eta_{1}=0.5$, (**b**) $\eta_{1}=0.2$, and (**c**) $\eta_{1}=0.05$.

**Figure 5 entropy-23-00073-f005:**
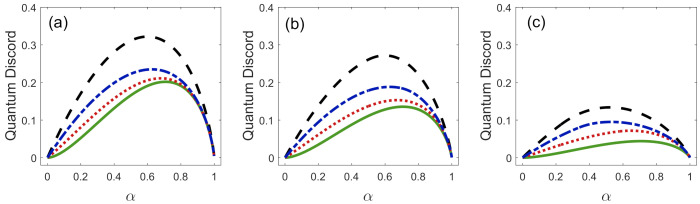
Quantum discord between Alice and Bob, $D(A|\left\{\Pi_{b}\right\})$, as a function of the overlap α in the discrimination by: ME where Q=0 (solid green line), FRIO with $Q=Q_{0}/3$ (dotted red line), FRIO with $Q=2Q_{0}/3$ (dashed-dotted blue line) and UD where $Q=Q_{max}$ (dashed black line) for: (**a**) $\eta_{1}=0.5$, (**b**) $\eta_{1}=0.2$ and (**c**) $\eta_{1}=0.05$.

## Data Availability

None.

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
