# Peer review of "Mutual Information and Quantum Discord in Quantum State Discrimination with a Fixed Rate of Inconclusive Outcomes"

_entropy, 2021, doi:10.3390/e23010073_

Round 1

Reviewer 1 Report

[Comment 1] A unified mathematical description of several detection strategies can be found in the literature by Nakahira et al.
[a] K. Nakahira, et al., ``Generalized quantum state discrimination problems," PRA 91, 052304 (2015). Refer this together with Ref.23

[Comment 2] Levitin's work for the accessible information is remarkable, but a limited result. Please not skip Fuchs's [b] and Ban's [c] works.

[b] Fuchs and Caves, ``Ensemble-Dependent Bounds for Accessible Information in Quantum Mechanics," PRL 73(23), 3047 (1994).
[c] Ban, et al., ``Upper bound of the accessible information and lower bound of the Bayes cost in quantum signal-detection processes," PRA 54(4), 2718 (1996). (Appendix:OPTIMIZATION OF THE BINARY QUANTUM DETECTION PROCESS)

[Comment 3] page 2, ``$\alpha=\lagnle \phi_1|\phi_2\rangle$"
In this paper, the parameter $\alpha$ is regarded as real.
Is this $\alpha=|\lagnle \phi_1|\phi_2\rangle|$ ?

[Comment 4] Some mathematical symbols are confusing:
Symbol $i$ in Equations (28)(29)(30); Symbol $\prime$ (') in Equations (35)(36)(37)(38)(42)(43)(49)(50) and others.

Author Response

Responses to Reviewer 1 Comments

We would like to thank Reviewer 1 for revising our manuscript and suggesting helpful changes in order to improve the presentation of the results from our study.

[Comment 1] A unified mathematical description of several detection strategies can be found in the literature by Nakahira et al.
[a] K. Nakahira, et al., ``Generalized quantum state discrimination problems," PRA 91, 052304 (2015). Refer this together with Ref.23

Response 1: As suggested by Reviewer 1, we added the aforementioned reference together with ref. Barnett, S.M.; Croke, S. Quantum State Discrimination. Adv. Opt. Photon. 2009, 1, 238–278, these references appear in line 35 of the new version of the manuscript.

[Comment 2] Levitin's work for the accessible information is remarkable, but a limited result. Please not skip Fuchs's [b] and Ban's [c] works.

[b] Fuchs and Caves, ``Ensemble-Dependent Bounds for Accessible Information in Quantum Mechanics," PRL 73(23), 3047 (1994).
[c] Ban, et al., ``Upper bound of the accessible information and lower bound of the Bayes cost in quantum signal-detection processes," PRA 54(4), 2718 (1996). (Appendix:OPTIMIZATION OF THE BINARY QUANTUM DETECTION PROCESS)

Response 2: The references [b] and [c], together with three other references on the topic of mutual information were added to the manuscript.

[Comment 3] page 2, ``$\alpha=\lagnle \phi_1|\phi_2\rangle$"
In this paper, the parameter $\alpha$ is regarded as real. Is this $\alpha=|\lagnle \phi_1|\phi_2\rangle|$ ?

Response 3: Yes, we considered the overlap as a real parameter. We added the following phrase, “The overlap between the non-orthogonal states is denoted by $\alpha=\langle\phi_1|\psi_2\rangle$, which we consider as a real parameter in the interval $\alpha\in [0, 1]$”. This appears in page 2 after line number 68.

[Comment 4] Some mathematical symbols are confusing:
Symbol $i$ in Equations (28)(29)(30); Symbol $\prime$ (') in Equations (35)(36)(37)(38)(42)(43)(49)(50) and others.

Response 4: We changed the notation in the entire manuscript from $|i’\rangle$ to $|\psi_i\rangle$. This change simplified the expressions in the new version of the manuscript.

Reviewer 2 Report

This work is about Classical and quantum correlations in quantum state discrimination with a fixed rate of inconclusive outcomes.
They studied correlations when quantum state discrimination with a fixed rate of inconclusive outcomes is performed.
However, many things should be clarified.

1. The motivation for this work should be clarified.
Why do the authors want to study correlations when quantum state discrimination with a fixed rate of inconclusive outcomes is performed?
Why do they need to study it?
2. There are many kinds of quantum correlations. However, in this work, the authors studied only discord. Why do the authors consider only discord for this work?
3. In the abstract, the authors insist, “The most (least) efficient discrimination scheme is ME (UD), from the point of view of correlations that are lost in the initial state and remain in the final state”.
How can the authors insist on this statement? As mentioned before, there are many types of quantum correlation.
4. In a clear way, the authors should present the result of the paper.

5.The authors used the new bases given eq(28),(29), and (30). How can these bases be justified for this work?

Author Response

Responses to Reviewer 2 Comments

We would like to thank Reviewer 2 for revising our manuscript and suggesting helpful changes in order to improve the presentation of the results from our study.

  1. The motivation for this work should be clarified.
    Why do the authors want to study correlations when quantum state discrimination with a fixed rate of inconclusive outcomes is performed? Why do they need to study it?

Response 1: One of the most important problems in quantum information theory is to quantify the mutual information that the sender and the receiver of a message share. In the present work, along with studying the discrimination with a fixed rate of inconclusive outcomes (FRIO), we also studied the mutual information that Alice and Bob share, and quantum discord or quantum correlations that are consumed in the process. This provides us with new knowledge that allows us a better understanding of the quantum state discrimination process. In particular, we studied FRIO discrimination because this scheme has two important particular cases in the protocols of quantum state discrimination, which are: minimum error discrimination (ME) and unambiguous state discrimination (UD).

  1. There are many kinds of quantum correlations. However, in this work, the authors studied only discord. Why do the authors consider only discord for this work?

Response 2: In general, quantum correlations can be used to evaluate how much disturbance a quantum measurement produces onto the joint initial quantum state. In particular, quantum discord can also be used for this task, as in this work. For instance, if the final joint state does not change after implementing the quantum measurement, then quantum discord is zero and the initial joint state has only classical correlations. In our work, we found two cases where quantum discord is zero, when the states {|\phi_1\rangle, |\phi_2\rangle} are orthogonal or when these states are equal. If quantum discord is different from zero, then the initial joint state changes under a quantum measurement process. This change is one of the characteristics of systems with quantum properties. Together with the aforementioned application, quantum discord is also related with the quantum correlations (measured by the quantum mutual information) that are lost in the measurement process. This topic is precisely one of the subjects of our work.

  1. In the abstract, the authors insist, “The most (least) efficient discrimination scheme is ME (UD), from the point of view of correlations that are lost in the initial state and remain in the final state”. How can the authors insist on this statement? As mentioned before, there are many types of quantum correlation.

Response 3: We are interested in the study of how information changes in a quantum state discrimination process. Our study focusses on the mutual information or the classical correlations that Alice and Bob share in the process of FRIO discrimination. Having evaluated the mutual information, quantum correlations in the form of quantum discord can be obtained by subtracting the mutual information to the quantum mutual information. In the case that Alice would encode and send information in two pure non-orthogonal states, and if Bob carried out a quantum measurement implementing ME, then the mutual information that he shares with Alice is maximized and, at the same time, quantum discord is minimized. If Bob implements UD instead, the situation is quite the opposite, i.e. , Alice and Bob will share the minimal mutual information and UD produces the biggest loss of quantum correlations or quantum discord.

  1. In a clear way, the authors should present the result of the paper.

Response 4: In order to clarify the motivation and results of this paper, we substantially modified: the title, the abstract, the introduction and the conclusions. The title changed from “Classical and quantum correlations in quantum state discrimination with fixed rate of inconclusive outcomes” to “Mutual information and quantum discord in quantum state discrimination with a fixed rate of inconclusive outcomes”. We think that the new title is more precise and informative than the previous one. Thus, we replaced the term “classical correlation” by “mutual information”, and the term “quantum correlation” by “quantum discord”.  In the introduction and the conclusions we explained: the aim of our work, the results that are already known from the literature, and our present contribution, in a form that is more precise and clear than the previous version of the manuscript.

  1. The authors used the new bases given eq(28),(29), and (30). How can these bases be justified for this work?

Response 5: In general, the FRIO discrimination process is carried out in a three dimensional Hilbert space. In order to find the classical correlations (maximal mutual information) or the optimum success probability in the discrimination, we need to find a basis for each of the mentioned cases. In general, these bases do not coincide. However, in our case they do coincide. We included this general basis because it is already known from the literature and it can be used to test our results in an analytical or numerical manner.